# Early Identification and Intervention in Pediatric Obsessive-Compulsive Disorder

**DOI:** 10.3390/brainsci13030399

**Published:** 2023-02-25

**Authors:** Xingyu Liu, Qing Fan

**Affiliations:** 1Shanghai Mental Health Center, Shanghai Jiao Tong University School of Medicine, Shanghai 200030, China; 2School of Psychology and Cognitive Science, East China Normal University, Shanghai 200062, China; 3Shanghai Key Laboratory of Psychotic Disorders, Shanghai 201108, China

**Keywords:** pediatric obsessive-compulsive disorder, prevention, early identification, early intervention

## Abstract

Obsessive-compulsive disorder (OCD) is a psychiatric disorder characterized by persistent thoughts with subsequent repetitive behaviors. Interventions that are effective for adult OCD cannot simply be generalized to pediatric OCD, since OCD in children and adolescents usually has a different clinical presentation, etiology and course from adult OCD. Delayed and inadequate treatment is associated with a worse prognosis, making the need for early identification and intervention in pediatric OCD very urgent. In this paper, we reflected on the current constraints that make early interventions for pediatric OCD unpromoted and reviewed the approaches with potential application for early identification and early intervention in pediatric OCD, categorized by three-level prevention stages corresponding to a clinical staging model. Since the etiology of pediatric OCD is still unclear, primary prevention is most lacking, and early interventions for pediatric OCD are currently focused on the secondary prevention stage, which aims to prevent the conversion of obsessive-compulsive symptoms into full-blown OCD; tertiary prevention mostly focuses on the alleviation of mild to moderate OCD, while interventions for co-morbidities are still in their infancy. We closed by considering the important research questions on this topic.

## 1. Introduction

Obsessive-compulsive disorder (OCD) is a psychiatric disorder characterized by intrusive, repeated and persistent thoughts, desires or images, with subsequent repetitive behaviors or thinking patterns that the individual performs in an attempt to decrease the anxiety or distress or simply according to rigid rules (i.e. compulsions) [1]. This chronic brain disorder demonstrates a bimodal onset, with one peak at 12–14 years and another at 20–22 years [2]. Studies in adults support the notion that OCD is a lifelong chronic disorder, whereas studies in youth suggest that a high percentage of patients have an episodic course [3]. As a distinctive subtype of OCD [4], pediatric-onset OCD affects around 2% to 4% of children and adolescents [5,6]. Considering that children and adolescents are in a critical stage of physical and cognitive development, the onset of OCD during this period may lead to additional developmental disruptions [7]. Symptoms of pediatric OCD may be less severe, but it can persist throughout the lifespan if left untreated, causing widespread academic, occupational and social damage and reducing the quality of life [8]. 

Considering that OCD is a debilitating disorder that worsens over time, and symptoms of the disorder can begin very early on—more than half of adult OCD patients have already exhibited subclinical symptoms in childhood [9,10]—there is an urgent need for early diagnosis and treatment [7,11,12,13], especially the strategies targeting the childhood age group [11]. Some studies have found that OCD remission rates in children are much higher and more persistent than in adults [3,8,14]. However, the duration of untreated obsessive-compulsive disorder remains one of the highest of all psychiatric disorders [15], with a gap between onset and diagnosis of approximately 7–10 years in adults and an average of more than two years in children [16,17]. These findings are worrying, especially because a longer disease course predicts a poorer long-term outcome [8]. Meanwhile, research on early identification and interventions for OCD is still in its infancy [7,13]. The OCD treatment guideline in the UK has emphasized the importance of stepped care, which aims to provide the most effective but least intrusive treatments appropriate to a person’s needs, but it did not indicate the concrete interventions that can be taken in the early stages [18]. Fineberg et al. [7] conceptualized the primary, secondary and tertiary prevention of OCD based on the clinical staging model of OCD proposed by Fontenelle and Yücel [19]; however, current research evidence of pediatric OCD is still very limited in constructing a complete prevention framework. 

In this article, we start with the specificities of pediatric OCD, with the goal of exploring why effective early intervention has yet to be fully implemented. Then, we make an attempt to complete the framework of the three-level prevention of pediatric OCD by carefully reviewing the current research evidence that helps identify at-risk populations and perform primary prevention for pediatric OCD and further assess the measures of the secondary and tertiary stages of early intervention that have currently been proven to work or may work. We close by identifying gaps in the existing literature and considering the important research questions on this topic.

## 2. Why Difficult?—A Perspective on the Characteristics of Pediatric OCD

As some of the specific disorder factors associated with OCD in youth are not common in adults, early interventions for pediatric OCD need to prescribe the right medicine for the right problem. Given the aim of this review, not all of the current clinical research evidence will be reviewed in detail; only those features that have an impact on early interventions for pediatric OCD will be included in this section.

### 2.1. Etiology: Diverse but Imprecise

To date, the pathogenesis and course of pediatric OCD remain unknown, with a likely combination of various genetic, neurobiological, environmental and psychological factors [20]. The unclear etiology makes it difficult for clinical scholars to target the precise direction of primary prevention for pediatric OCD.

The heritability of pediatric OCD yields higher genetic loading than adult OCD [21]. Family and twin studies, heritability assessments, genetic linkage, candidate gene association studies, genome-wide association studies (GWAS), epigenetic studies and genome sequencing all provide strong evidence for a genetic basis for OCD [21,22,23,24,25,26]. Animal models also have much to provide concerning the analysis of possible factors in etiology [27]. Many candidate genes have been identified in recent years, a significant proportion of which are related to the glutamatergic system, but finding suitable molecular therapeutic targets is still challenging [28]. The lack of consistency and confidence also reduces the application potential of these studies [13,25,29]. Achieving more personalized interventions requires seeking genetic evidence that may address the phenotypic and treatment heterogeneity of OCD [28].

From the neurobiological perspective, the cortical-striatal-thalamic circuits (CSTC) are the most implicated in pediatric OCD [30,31]. Several models of how these fronto-striatal loops are impaired have been put forward to explain OCD symptoms [27]. The different activity and volume of the Anterior Cingulate Cortex, lateral and medial Orbitofrontal Cortex, Basal Ganglia and Corpus Callosum are also found between healthy controls and those with OCD in pediatric samples [32]. Neuroimaging methods can help us find indicators applicable to the early identification of pediatric OCD. For example, Boedhoe et al. [33] conducted a meta-analysis and a mega-analysis with T1-weighted MRI images of 1830 OCD patients and 1759 control participants from datasets of the ENIGMA OCD Working Group, indicating that the pallidum and hippocampus seem to be more critical in adult OCD. In contrast, the thalamus seems to be key in pediatric OCD. They suggested that an increased thalamic volume may be an early sign of pediatric OCD and may be related to altered neurodevelopment. Another study underscored the importance of parietal regions in pediatric OCD, which also found more pronounced surface area deficits (mainly in frontal regions) in medicated pediatric OCD patients [34]. The reported heterogeneity between pediatric and adult neuroimaging findings suggests that the development of OCD might be nonlinear, which means early intervention for pediatric OCD requires specialization. However, it cannot be confirmed based on cross-sectional studies alone, and longitudinal studies in neuroimaging are badly needed [27]. A final point to add is that one study has identified similar brain loops between OCS and OCD in children, indicating that neuroimaging studies have the potential to provide evidence for the efficacy of early intervention for pediatric OCD [35]. 

The validity of the autoimmune and neuroinflammatory hypothesis of pediatric OCD remains to be tested. Initial supportive evidence derived from the models of Sydenham’s Chorea [36] and Pediatric Autoimmune Neuropsychiatric Disorder Associated with Streptococcus (PANDAS) [37], which suggest an overactive autoimmune response in group A Beta-Hemolytic Streptococcus (GABHS) infections can lead to obsessive-compulsive symptoms. Later, the immune component of pediatric OCD has been identified in many studies, including associations with low-grade inflammation, neuro-inflammation, autoimmune disease and neuro antibodies (especially autoantibodies against the basal ganglia) [38,39,40,41,42,43,44,45]. In recent years, translocator protein (TSPO) positron emission tomography imaging has provided a direct brain measure of inflammation. By using this technique, researchers found greater TSPO binding within the dorsal caudate, orbitofrontal cortex, thalamus, ventral striatum and dorsal putamen of 20 OCD patients compared to matched healthy controls, indicating that the aberrant inflammatory processes in OCD might extend beyond the basal ganglia to include the cortical-striatal-thalamic circuits (CSTC) [46,47]. However, the time frame between a given streptococcal infection event and the development of OCD remains to be clarified [41,42,48]. A large epidemiological study (*N* = 678,862) showed that patients diagnosed with OCD did not actually have a higher risk of prior possible streptococcal infection [42]. Moreover, a meta-analysis including 538 OCD cases and 439 healthy controls showed that patients with OCD do not exhibit increased peripheral inflammation during the course of the disease, as no consistent differences in representative biomarkers such as TNF-α, IL-6, IL-1β, IL-4, IL-10 and interferon-γ were found between the two groups [49]. In general, immune abnormalities cannot yet be explained as a stable and common cause of pediatric OCD. Some scholars have suggested that “autoimmune OCD” should be distinguished as a separate subtype, requiring the establishment of specific diagnostic criteria to distinguish cases of secondary immune-mediate OCD that account for the minority of the overall population of OCD patients [45]. Based on current findings, it is difficult to justify considering immunotherapy at the primary stage of intervention for pediatric OCD, but the introduction of the treatment of pediatric acute-onset neuropsychiatric syndrome (PANAS) [43,50] to reduce the inflammatory burden of OCD patients at tertiary prevention could be an attractive option. Given that the generality of immunomodulatory interventions and of the biomarkers that can predict their efficacy remains unclear [38,43,45,49,51,52], we will not discuss in detail the role of immune reflection in secondary prevention in Section 3.

As for environmental and psychological factors, although some of them were found to play a role in the etiology of pediatric OCD, no causality has been shown for any of the risk factors studied [53,54,55]. However, the interaction between environment and genes may be related to the causes of pediatric OCD, as higher familial aggregation [29] and epigenetic influences [56,57] are observed in early-onset OCD. Therefore, intervention in environmental factors (e.g., improving the environment that contributes to the expression of genes associated with pediatric OCD) is an essential direction for primary prevention [7]. In addition, family is an important environmental factor that influences the course and prognosis of OCD in children and adolescents; thus, it is necessary to treat patients and their families as a whole in the early treatment of pediatric OCD. In the next section, we will elaborate on what existing early intervention approaches have done for environmental factors.

### 2.2. Clinical Descriptions: Normal or Abnormal?

Corresponding to the Yale-Brown obsessive–compulsive scale (Y-BOCS) [58,59] for adults, the children’s Yale-Brown obsessive–compulsive scale (CY-BOCS) is the most popular and widely used measure of pediatric OCD [60,61,62,63]. Obsession and compulsion severity are rated on distress, frequency, interference, resistance and symptom control.

In order to effectively identify individuals in need of secondary prevention, it is important to identify the prodrome and trajectory of pediatric OCD from existing research. We will not meticulously discuss the characteristics of pediatric OCD in each symptom dimension since clinical studies do not paint a consistent picture of them [64,65,66]. What is of concern is to distinguish between normal and pathological situations, given that young children often exhibit ritualized behaviors (e.g., bedtime routines and mealtime routines) or beliefs (e.g., superstition and insistence on sameness) that are developmentally appropriate but may reflect obsessive-compulsive symptoms (OCS) [67,68,69]. Currently, normative rituals/routines and OCS are widely believed to exist on a continuum with shared emotional, neurobiological and genetic factors, and age plays an essential role in determining whether rituals/routines are normal or maladaptive [68]. Children’s ritualistic behavior typically starts to decrease from age 7 [70], and children who consistently perform high levels of rituals/routines are at risk for developing OCS [68]. According to the analysis result of data from a longitudinal study [69], OCS trajectories can be divided into three distinct groups. Overall, the majority of children in the study fell within the No Peak group with low or no OCS levels at all time points. Other children can be divided into two diametrically opposite categories: a progressive worsening or reduction in symptoms from preschool to high school. Therefore, examining the presence of OCS in early childhood may not be adequate to differentiate children who display normal levels of ritualistic behavior from those who may experience heightened OCS in later life [68], nor does it help us identify those in need of early intervention accurately. To address this issue, exploring multiple variables is crucial for identifying at-risk populations [32], the current progress of which will be reviewed in detail later. 

Another concern that needs to be mentioned is that while the common hypothesis regards compulsions being developed to reduce anxiety caused by obsessive thoughts, the necessity for this criterion in pediatric OCD diagnosis needs to be reconsidered. Children are likely unaware of why they engage in compulsive behaviors, and some deny that their compulsions are motivated by obsession [71,72]. Furthermore, although OCD developed during adolescence and after adolescence is usually associated with anxiety and depression, this does not hold true in childhood-onset OCD [66]. These facts not only pose a considerable challenge to the diagnosis of pediatric OCD but also strongly challenge the causal relationship between obsessive and compulsive symptoms [28,64].

### 2.3. Comorbidity: Intricate and Complex 

Comorbidity often predicts a more complex course of OCD and poorer treatment outcomes [1], also consistently related to the worsening symptom severity of subclinical OCD [73,74]. Co-occurring psychopathology implies the need for transdiagnostic identification and intervention at an early stage of tertiary prevention. 

The trinity of ADHD, OCD and tic syndrome is often found in the child and adolescent population [75,76,77]. Unlike adult OCD, pediatric-onset OCD is slightly more common in boys than in girls, and an increased prevalence of comorbidity with attention deficit hyperactivity disorder (ADHD) and tic disorders is seen in these boys [16]. Many studies have explored common risk factors across these three disorders [78,79], but to our knowledge effective early intervention protocols for such comorbidities are not yet available.

The comorbidities of pediatric OCD with schizophrenia and bipolar disorder are also of concern. Pediatric OCD usually precedes these two mental disorders and might be an early warning flag for their possible later onset [80,81]. Recently, Preti et al. [82] have proposed a strategy for early detection and intervention based on the co-occurring symptoms of OCD and these two psychiatric disorders, but specific protocols and empirical testing are still lacking.

Other Common comorbidities include major depression, anxiety disorders, disruptive behavior disorders [12,64,83], etc.; transdiagnostic emotion-focused treatment for these comorbidities has shown to be potentially effective, as detailed in Section 3.

Overall, we still lack sufficient cross-diagnostic studies (based on genetic, neurobiological and behavioral approaches, etc.) to help us better understand the mechanism of OCD, the identify markers of risk, and provide opportunities to develop specialized interventions.

### 2.4. Insight of Patients: Insensitive and Naive

As mentioned above, although the treatment latency for pediatric OCD is considerably shorter than that for adult OCD, two years is still a long delay if we take into account that the average age of those youths diagnosed with pediatric OCD is only 13 [17]. In addition to the limitations of clinical evidence, there are also factors in the patients themselves that prevent them from getting early intervention.

The most significant barrier to early intervention for children and adolescents is that they may not know they need it. Compared to adults, children are less able to articulate their obsessional thoughts, often lack the mature insight to recognize that their compulsive psychological experiences are unreasonable [13,16] and lack the vocabulary and comprehension to express what is happening to them. Moreover, children may not be able to separate reality from fantasy, making it difficult for them to question the validity of the connection between their thoughts and behaviors. They can sincerely believe that their rituals are necessary [84]. In other cases, younger children showed simply subclinical OCD symptoms but did not meet all the diagnostic criteria for full-blown OCD. They do not personally experience anxiety or distress about engaging in repetitive behaviors but only show irritation when interrupted or interfered with. Even those familiar with these symptoms may be unaware of the role that the brain plays in OCD and may mistakenly believe they can solve the problem [8,84]. A lack of awareness about pediatric OCD and understanding of its neurological basis are the two main obstacles to timely diagnosis and treatment. 

Another factor deterring children from seeking help for OCD is that they may be ashamed of their symptoms or afraid of seeming crazy. Those obsessive thoughts of sex or hurting someone in their mind can make them feel guilty and prevent children from sharing them with anyone who might reach out.

In summary, the unique nature of pediatric OCD and the cognitive limitations of children and adolescents remind us that simply applying the treatment of adult OCD to pediatric OCD individuals may not work. Systematic identification, prevention and intervention for pediatric OCD require specialized research and knowledge to guide, which will be reviewed in detail in the following sections. 

## 3. Primary, Secondary and Tertiary Prevention of Pediatric OCD 

Applying the concept of the three-level prevention to early intervention for OCD has been called for by many scholars [7,11,13]. Primary prevention aims at preventing the development of OCS. Secondary prevention is identifying and treating individuals who develop OCS to prevent progression to full-blown OCD. Finally, tertiary prevention reduces the significant comorbidity and minimizes the impact of OCD once individuals have been diagnosed [7,11]. At present, early interventions for OCD achieved in the clinical field are focused on the stage of secondary prevention. However, in this paper we suggest that all three prevention levels should be taken into account. We will then separately discuss the implementation of each level of prevention in pediatric OCD.

### 3.1. Primary Prevention

The achievement of primary prevention requires, at least, the precise identification of at-risk individuals for pediatric OCD and the implementation of a specific approach to improve their resilience and resistance to OCD symptoms developing. However, limitations in understanding the etiology of pediatric OCD have led to very little progress in either aspect. In the following, we will review the new approaches that have emerged in recent years that may help advance the identification of at-risk populations and provide an outlook on advisable directions for primary prevention.

#### 3.1.1. Polygenic Risk Scores (PRS)

Although existing genetic studies cannot yet help us determine the exact etiology of pediatric OCD, it may be feasible to identify at-risk groups with the help of Polygenic Risk Scores (PRS). Based on GWAS findings, PRS reveals an individual’s genetic risk of developing a disease by detecting all the risk alleles carried by the individual and weighting them according to their effects [85]. 

It has been found that the OCD PRS can significantly predict an individual’s OCD status, and the personality trait of harm avoidance mediates the association between OCD PRS and OCD diagnosis [86]. Similarly, the potential of the application of PRS in identifying subclinical populations of pediatric OCD was also demonstrated in another study, as it was discovered that polygenic risk for OC traits was associated with OCD case/control status and vice versa [87]. However, the cross-race consistency of PRS results remains to be tested, and the large-scale clinical adoption of PRS for risk assessment and disease diagnosis is still not achievable [13].

One study also investigated whether PRS could be used to predict treatment outcomes in OCD, although the outcome could not demonstrate its practicality [88]. This result needs to be verified by more investigation.

#### 3.1.2. Identifying Endophenotypes 

Endophenotypes are inheritable intrinsic traits between the genes and the extrinsic phenotype. They can be found both in patients and their unaffected relatives [89], and need to be measured by neurobiological, biochemical, neuroanatomical, cognitive and neuropsychological laboratory techniques [90]. Because endophenotypes are more closely related to the underlying genetic basis than the behavioral phenotypes and are less influenced by genetic and environmental heterogeneity than the disease itself, they are thought to be more helpful in identifying the genetic variants, etiology and the pathophysiological basis of the disease [89]. By identifying the endophenotypes of pediatric OCD, we can further precisely locate the genes affecting those endophenotypes and thus learn about the biological pathways affecting disease susceptibility, which could help us to identify populations at high risk for pediatric OCD and inform potential targets for early intervention [28,89,91].

A systematic review focused on the currently proposed neurocognitive endophenotypes of pediatric OCD in 43 studies, finding that abnormal action monitoring is considered a robust endophenotypic feature of pediatric OCD, which has been confirmed by high amplitudes of ERN and abnormal activation in ACC; intolerance of uncertainty, possible impairment of planning ability and the hyperactivity of the frontoparietal regions in working memory tasks are potential endophenotypes, though the results of the available studies are inconsistent [92]. A recent study has provided insights into error-related brain activity in pediatric OCD by time-domain and time-frequency analysis, suggesting that pediatric OCD may be characterized by enhanced error monitoring (i.e., greater theta power) and post-error inhibition (i.e., greater beta power) [93]. However, the participants selected for the study were only pediatric OCD patients, which means the current result has not been validated in their unaffected relatives. Therefore, it remains to be tested whether this can be considered a neurobiological marker for screening at-risk populations.

In addition, deficits in cognitive flexibility and response inhibition, which are commonly found in adult OCD, have not been demonstrated as neurocognitive at-risk markers in pediatric OCD [92,94], suggesting that the cognitive function of our pediatric OCD patients may not be impaired at the onset, but will gradually lead from neurological dysfunction to cognitive impairment during the course of disease progression [95]. Therefore, more attention should be paid to neurological dysfunction rather than abnormalities in cognitive-behavioral performance.

Linking candidate neuropsychological endophenotypes to single nucleotide polymorphisms (SNPs) in genotypes could help us to confirm whether these intrinsic traits are associated with any candidate genes for OCD and, thus, how endophenotypes interact with genes, environment and disease [92]. A recent study has combined neuroimaging analysis, GWAS and PRS to investigate genetic variants associated with pediatric obsessive-compulsive behaviors (OCB) and imaging endophenotypes, meanwhile determining the relationship between brain activity and pediatric OCB. However, no significant results have been found so far [96]. More research is needed in this field.

Besides, it is now widely recognized that psychiatric disorders are heterogeneous; thus, not everyone with a disorder can be expected to have the same endophenotype, nor are endophenotypes necessarily disorder-specific [89], which is vital for understanding the genetic basis of comorbidities and may contribute to tertiary prevention, yet practice evidence is lacking. 

#### 3.1.3. Psychoeducation and Life Style Interventions

Given that the target population for primary prevention has not yet shown typical OCS, the intervention approach for them should be different from that for patients with psychiatric disorders. 

The relatively viable natural candidate is psychoeducation. Psychological education in primary interventions is expected to raise people’s awareness of pediatric OCD and help parents identify their children’s risk traits early [11,19,97]. However, these strategies have not been fully evaluated, and we have no idea whether psychoeducation is likely to be effective in protecting against pediatric OCD.

Another helpful attempt is lifestyle intervention. Potential intervention targets supported by current research evidence are developing anti-inflammatory diets and increasing physical activity [98]. Other researchers have suggested that interventions should focus on reducing stress and improving sleep quality [19]. These proposals have also yet to be empirically tested.

Finally, as mentioned before, the difficulty in implementing primary prevention lies not only in our limited knowledge of the etiology of pediatric OCD but also in the fact that at-risk individuals without clinical symptoms are not motivated to seek help. Therefore, the scientific awareness of pediatric OCD should be widespread in the general population, requiring numerous professionals’ involvement.

Overall, psychoeducation is currently the most viable intervention in the primary prevention stage. We believe the involvement of schools will contribute to spreading its impact. For example, providing educational lectures and courses at all grade levels in school to enable more students and their parents to tell if they are at high risk (e.g., having a family history of OCD) may help bridge the target population and other prevention strategies that are appropriate for them. We also consider introducing neuropsychological tests that do not require complex equipment into the routine health examination program for all populations as a cost-effective way to help with early identification, which also has great potential for future applications. As for PRS and neuroimaging, which require the support of complex equipment and technical expertise, it is difficult to achieve population-wide application. When they do become available for clinical use, making these techniques available to children and adolescents with a family history of OCD is the primary goal we believe we should strive to achieve, although this process still faces the challenges of ethics, health economics and the availability of resources to action, etc. [99].

### 3.2. Secondary Prevention

In the secondary prevention phase, it is necessary to identify patients with obsessive-compulsive symptoms early and to enable them to be treated as soon as possible to avoid the development of full-blown OCD. Current identification methods, as well as interventions that contribute to secondary prevention, will be reviewed in this section.

#### 3.2.1. Screening for Early Symptoms 

Clinicians should monitor obsessive-compulsive symptoms in children and adolescents, even if they are not currently perceived or complained about by the subject [100]. The school should also be actively involved in this process, considering that students spend so much time on campus. Nevertheless, the efforts of school psychologists in this area have been minimal [101,102]. Several studies have identified potential subthreshold/early symptoms of OCD. For example, symptoms of the symmetry and order dimensions (detected by the Dimensional Yale-Brown Obsessive-Compulsive Scale, DY-BOCS [103]) that have been found to have the earliest onset (13.6 ± 8.6) [104]; “Can’t get mind of certain thoughts” and “Fears might think or do something bad” are the most common symptoms among those who exhibit high OCS in early childhood [69], while “Repeats certain acts over and over” is the most common symptom among those who exhibit high OCS in adolescent periods (detected by the eight-item Obsessive-Compulsive Scale [105]). As mentioned in Section 2.2, a precise distinction needs to be made between normal rituals/routines and maladaptive OCS in the evaluating process, and age is currently the most helpful basis for clinicians to make judgments [68].

Not all early symptoms will turn into full-blown OCD [106]. Based on current knowledge, children and adolescents with OCS are more likely to develop full-blown OCD if they present with other risk factors such as parents with OCD [107,108], maladaptive parenting style [107], attention problems [69], anxiety and depression disorder [74,109] at the same time. Therefore, the abovementioned factors must be carefully investigated while monitoring for OCS.

Long-term follow-up testing is recommended, with immediate intervention if symptoms continue to deteriorate.

#### 3.2.2. Bibliotherapy

Bibliotherapy is the use of books or stories for therapeutic purposes to help an individual gain insight into his or her problems [84]. It not only helps children to understand complex concepts better and cope with frustrating news but also helps clinicians and parents who have difficulty describing the nature of a diagnosis of a disease to their children [84]. All kinds of books can be used for bibliotherapy, including poetry, storytelling, fiction or nonfiction [110]. It can educate readers, provide insights, encourage discussion, provide solutions to problems or new ways of looking at them and make people realize that they are not alone in their situation [111]. 

Attempts have been made to use bibliotherapy for the secondary prevention of pediatric OCD. *Amazing Adam and the Secret Spell* is an illustrated book written for children ages 6-10 who are experiencing symptoms of OCD, designed to help readers identify and understand their obsessive-compulsive symptoms, overcome shame and encourage them to seek treatment [84]. So far, readers’ feedback has proven its promising application, yet the large-scale dissemination and output of more language versions of the same type of picture book are still lacking.

Another use of bibliotherapy in pediatric OCD early intervention is showing children how to process self-help cognitive-behavioral therapy (CBT). Cognitive-behavioral therapy is a first-line treatment for OCD in children and adolescents in line with a robust evidence base [100,112]. Some examples of CBT that individuals can implement include activity scheduling, goal setting and cognitive restructuring. Until now, there have been at least three CBT self-help books written specifically for children and adolescents with OCD [113,114,115], but studies of the effectiveness of such books as an intervention in clinical settings are scarce; therefore, their effectiveness and applicability need to be further confirmed [112]. 

#### 3.2.3. Novel Digital Interventions

The popularity of smartphones and smart electronic wearables provides new opportunities for the early identification of intervention in pediatric OCD. 

First, the development of digital phenotypes has led to more observable indicators for the early identification of obsessive-compulsive disorder. For example, some forms of the problematic use of the Internet (PUI) associated with OCD (e.g., repeatedly checking social media, digital hoarding, etc.) can be accurately recorded by digital media. In turn, clinicians can use big data analysis to identify individuals who exhibit digital obsessive-compulsive symptoms [27,116,117]. 

At the same time, the web-based self-assessment questionnaire also showed good sensitivity and sufficient specificity for detecting OCS when compared with a full-length structured clinical interview. This form of measurement can effectively make it easier for more patients and at-risk individuals to identify and monitor their mental health status [116].

Furthermore, a smartphone app for conducting CBT interventions for OCD has been proven effective, allowing patients to perform relevant CBT exercises at home to recognize and improve their emotions and maladaptive cognitions, helping to reduce OCD-related beliefs and symptoms [118].

Finally, the proactive use of webcams and smartphone cameras gives clinicians the opportunity to monitor individuals’ behavior in the natural environment and the possibility to intervene remotely with CBT, which can significantly reduce the cost of disorder treatment and achieve early detection and intervention for secondary prevention [117].

A professional team has developed an enhanced cognitive behavioral therapy (eCBT) package for pediatric OCD, including a smartphone app with psychoeducational tools, OCD-related symptom assessment tools and ERP training guidance tools for pediatric OCD patients and their parents. It also contains a video conferencing platform that supports face-to-face therapy sessions using a webcam. The effectiveness of this package has been preliminarily proven to have positive treatment outcomes, and it is expected to see more integrated online intervention products promoted in the population in the future [119,120].

### 3.3. Tertiary Prevention

To date, the duration of illness before treatment was the only stable predictor of long-term course in pediatric OCD [3,121]. Evidence from multiple studies suggests that longer untreated illness duration is associated with many unwanted outcomes, including higher rates of comorbidity and disability, greater family accommodation [122,123], poorer treatment response [124,125,126] and more delayed [127] and less frequent remissions [128]. Therefore, clinicians should provide patients with treatment as early as possible after symptoms have reached the severity of a clinical diagnosis of OCD. According to existing studies, it is not confirmed whether factors such as the severity of symptoms, family accommodation, family history of OCD and comorbidities are associated with a long-term course [3,121], but they are usually related to treatment response and symptom/diagnostic remission [129,130,131]. Improving existing evidence-based treatment protocols to enable patients to receive adequate, specific treatment early is essential to enhance tertiary prevention. We will elaborate on current options for the potentially more effective evidence-based treatment of pediatric OCD in the following. 

#### 3.3.1. Pharmacotherapy and Psychotherapy: Alone or in Combination

Effective pharmacological and psychological treatments for pediatric OCD have been established and in clinical use for many years [132]. Among the pharmacological treatments, selective serotonin reuptake inhibitors (SSRIs) are considered the first-line option for pediatric OCD patients [133,134], including fluoxetine, fluvoxamine, sertraline, paroxetine and escitalopram, etc. Patients may respond differently to various medications, and in cases of non-response or inadequate response to one SSRI, another SSRI should be tried [117]. Psychotherapies such as cognitive behavioral therapy (CBT), especially with exposure and response prevention (ERP), are also recommended treatments for children and adolescents with OCD [100,135]. 

A network meta-analysis showed that the effects of accepting CBT alone and combined with drug therapy were comparable and better than those for youths with mild and severe pediatric OCD [136]. Recently, a retrospective study also showed that one-third of the children and adolescents in the sample did not respond to treatment with SRIs alone, and the existence of symmetry/hoarding symptoms is also associated with poorer response to pharmacological treatments [137]. However, not all patients can benefit from a single CBT treatment in the early treatment phase; for children and adolescents with a family history of OCD, only CBT combined with SSRIs treatment may result in significant improvement [130]. 

Evidence suggests that medication is not preferred for the early treatment of pediatric OCD compared to CBT. When making specific clinical decisions, information on the patient’s characteristics, such as the family history of OCD and clinical presentation of symptoms, must be considered to optimize treatment planning and provide more accurate prognostic information. More longitudinal studies are required to explore which treatment options are more resistant in reducing relapse and chronic functional disability in the long term.

#### 3.3.2. Family-Based Treatment 

Considering that children are deeply embedded in family units, the prospects for intervention in the family environment in tertiary prevention are of great concern.

One of the most well-studied structures of family factors in pediatric OCD is family accommodation (FA), which refers to the act of parents, siblings, or partners accommodating the high-risk individual’s requests to comply with their compulsions in order to avoid or alleviate distress. It is associated with a greater severity of OCD symptoms, more impairment and poorer treatment outcomes [138,139,140]. Reducing FA is considered a promising intervention in all three levels of prevention. However, in this review we only mention it in the third stage because the impact of FA on full-blown OCD has been most intensively studied, while evidence from longitudinal studies that could demonstrate the efficiency of interventions for FA in the primary and secondary prevention stages of pediatric OCD is still lacking [7,11]. 

Related to the increase in FA are greater parent–child conflict and parental blame. It takes time and effort for parents to adjust to their child’s compulsive behaviors; therefore, a child’s compulsion or avoidance is not tolerated every time. When a parent refuses to accept the behavior of their child with pediatric OCD, the child may show anger or behavioral outbursts [141,142]; in addition, the parent may feel guilty or have more blame expressed toward the patient [20,143]. All of these can affect treatment outcomes [144]. Therefore, the role of family-based intervention approaches for pediatric OCD cannot be underestimated. 

There are many ways that parents can be involved in their child’s OCD treatment process. Psychoeducation is one of the available forms; it can be conducted for parents alone or as part of family-based CBT (FCBT) treatment for different purposes, such as increasing parents’ knowledge of mental health, helping create a more supportive family environment, improving their parenting styles and raising parents’ awareness of FA [20]. Furthermore, FCBT interventions include teaching parents to reduce FA, training them to assist their children with homework and ERP therapy and helping parents identify and cope with the pain and anxiety they feel when facing their child’s OCD [145,146,147,148]. Studies have shown that these FCBT techniques have similar effects on parents and OCD children [145]. Parents should also learn to properly communicate with their children about thoughts and emotions [20]. Group FCBT has also provided a new approach to parental involvement, although the current evidence suggests that it is not as effective as individual FCBT [145]. To our knowledge, no current studies have compared the efficacy of FCBT with conventional CBT for pediatric OCD, but given the high familial heritability of pediatric OCD [29], promoting family-based treatment is a necessary complement to traditional first-line psychotherapy.

#### 3.3.3. Transdiagnostic Treatment 

Although the DSM-5 constructed classification system for mental disorders is dominant in clinical practice, which has detailed diagnostic criteria for OCD, it also acknowledges that the symptoms in the current diagnostic criteria are not specific to OCD alone. There are 14 other disorders (or disorder classes) whose symptoms also broadly fit these criteria, and even after a careful differential diagnosis, co-morbidities remain in many cases [149,150]. As a result, there are calls for the creation of new biologically relevant transdiagnostic traits linking genes, molecules, cells, neural circuits and behavioral manifestations across multiple diseases [149]. The therapeutic approaches thus extended will focus on the link between neurocognitive markers and treatment response. 

In pharmacotherapy, some studies have begun to investigate whether neurological markers (e.g., OFC gray-matter volume) shared by OCD with other mental disorders can be used as predictors of the treatment response for SSRIs [151] in order to assist in determining the optimal medication dose when coping with co-morbidities in clinical decision making, though current studies have not obtained stable results at present, and study samples in children and adolescent populations are lacking. 

As for psychotherapy, principle-driven intervention protocols that target underlying shared mechanisms of comorbidities rather than symptoms complement evidence-based treatment [152]. An emotion-focused transdiagnostic intervention therapy, the Unified Protocols for Transdiagnostic Treatment of Emotional Disorders in Children and Adolescents (UP-C/UP-A) [153], has been shown with significant improvement in obsessive-compulsive symptoms in children and adolescents. However, the samples selected for this study were all patients with anxiety/depression disorders; it is unclear whether equivalent effects would be seen in patients with OCD co-morbidities [154].

In addition, the potential application of a non-invasive neurostimulation technique, transcutaneous auricular vagus nerve stimulation (taVNS), is of interest to treat neurodevelopmental disorders in children and adolescents. It has been shown that by targeting stimulation sites and parameters, taVNS has modulatory effects on cortical and subcortical brain regions associated with the neuropathology of ADHD, DBD, ASD and OCD, etc., further helping regulate some impaired social-emotional functions that are impaired in them [155,156,157,158]. This technique has an extensive potential application for reducing functional impairment in the tertiary prevention phase of pediatric OCD and deserves to be studied on a larger scale in the pediatric population.

## 4. Discussion

The demand for early identification and intervention in OCD has persisted for many years; however, systematic and long-term practical approaches are still limited, especially in pediatric OCD. This is mainly because the diagnosis of pediatric OCD is more difficult, the comorbidity is more specific and the etiology of pediatric OCD presents differences from that of adult OCD, so those interventions for adults cannot be simply applied to children and adolescents. In this article, we review approaches with potential application for early identification and early intervention in pediatric OCD, categorized by three-level prevention stages corresponding to the clinical staging model of OCD proposed by Fontenelle and Yücel [19]. The general pattern is shown in Table 1.

Overall, there is little research evidence for primary prevention, and prospective randomized controlled trials are badly needed to validate the identification of high-risk populations and early intervention. The secondary prevention phase is relatively well established but has yet to be widely promoted, especially in cross-cultural contexts. Tertiary preventions are improvements on traditional psychological and pharmacological treatments, with more promising results in terms of proving their effectiveness, but intervention for comorbidities is still in its infancy. All three stages of intervention approaches lack prospective longitudinal controlled studies of child and adolescent populations to demonstrate long-term impact (e.g., the reduction of morbidity, progression of symptoms, or retention of children’s ability to reach developmental milestones).

In order to advance effective early identification and intervention in pediatric OCD, the following questions await adequate empirical research in the future. 

First and foremost, more attention needs to be paid to pathogenesis, which affects the possibility of early identification and is crucial for selecting indicators for the effectiveness of treatment. Because of more new neuroscience tools being developed and the relatively low cost of exome sequencing, we can now test for causality by manipulating neural loops in animal models. Mechanistic studies in animal models may provide information about potential new therapeutic targets for OCD [160]. The habit hypothesis also provides important clinical insights about effectively interrupting the reinforcement cycle of OCD [7,117], which may contribute to preventing OCS from converting to full-blown OCD. It is of equal importance to study the natural course of pediatric OCD, as it may help identify at-risk children and adolescents early. As mentioned above (see Section 2.2 for more details), Luke et al. have set an excellent example [69]. Furthermore, we still need more information about factors that can influence the development of the severity and amount of OCS; prospective longitudinal studies on this topic will be precious in helping us to identify populations that are in greater need of benefit from psychological education and symptom observation in early life [32]. Attention should be drawn to the fact that all findings shown to be robust in adult OCD require re-testing in the child and adolescent population to verify their applicability in pediatric OCD.

Second, the potential application of non-invasive neurostimulation techniques (such as rTMS and tDCS) in pediatric OCD also deserves attention. Such techniques are widely used in treating OCD in adults [27,117] but have not been adopted for pediatric OCD interventions. Since such techniques have specific targets for stimulation and can directly modulate the activity of the corresponding brain regions, they may be very promising in the field of transdiagnostic interventions for pediatrics. The most effective stimulation targets selected for rTMS in adult OCD, such as DLPFC, OFC, SMA and mPFC-ACC, were all determined based on consistent results obtained from numerous previous neuroimaging studies [161,162]. In light of that, further neuroimaging studies in pediatrics remain necessary. Existing results are insufficient to help us pinpoint which brain regions are critical to improving OCD symptoms in children and adolescents through non-invasive neurostimulation. Meta- and mega-analysis based on worldwide data from existing studies would also significantly contribute to this goal, which calls for a global collaboration [163].

Furthermore, all the early intervention programs should be tested for cross-cultural adaptability. Whether the findings obtained in western developed countries with white populations are still valid in other corners of the world and whether available intervention approaches can benefit people of different cultural backgrounds and social statuses are matters for researchers and clinical psychologists to consider.

## 5. Conclusions

Early identification and intervention for pediatric OCD require specific theoretical guidance and research evidence to support them. The proposal of hierarchical prevention strategies guided by a clinical staging model is promising; however, it requires greater investment in research on the pathogenesis of pediatric OCD as well as the long-term and large-scale effectiveness of established early intervention approaches.

## Figures and Tables

**Table 1 brainsci-13-00399-t001:** Primary, secondary and tertiary prevention corresponding to the clinical staging models of pediatric OCD ^1^.

OCD Staging Levels		Symptoms ^2^ (CY-BOCS Scores)	Risk Factors	Prevention Levels	Identification	Proposed Intervention
Stage 0		NA ^3^(0)		Primary prevention		Watchful observation ^4^ *, psychoeducation, lifestyle intervention
	Stage 0A		Family history of OCD or neural vulnerabilities		Polygenic risk scores (PRS), neuroimaging, neuropsychological tasks	
	Stage 0B		Environmental risk factors *		NA	
	Stage 0AB		Family history of OCD or neural vulnerabilities AND environmental risk factors *		Polygenic risk scores (PRS), neuroimaging, neuropsychological tasks	
Stage 1		Subthreshold (1–13)	Family history of OCD or neural vulnerabilities AND/OR environmental risk factors	Secondary prevention	DSM-5, CY-BOCS, neuroimaging, neuropsychological tasks	cognitive behavioral workshops, bibliotherapy, self-support digital interventions
Stage 2		Mild to moderate (14–30)	Variable	Tertiary prevention	DSM-5, CY-BOCS	CBT and/or SSRIs, parental involvement treatments, non-invasive neurostimulation
	Stage 2A ^5^					
	Stage 2B ^5^					
Stage 3		Severe (31–40)	Variable	NA	DSM-5, CY-BOCS	
	Stage 3A					SSRIs and CBT, non-invasive neurostimulation
	Stage 3B					SSRIs and CBT, invasive neurostimulation, psychiatric surgery

^1^ This table refers to the OCD staging model proposed by Fontenelle and Yücel [19]. ^2^ The severity of symptoms was classified according to Lewin et al. [159]. ^3^ NA = not available; ^4^ * = no evidence supporting efficacy as of now. ^5^ The substage level of stage 2 represents different courses of pediatric OCD. Stage 2A corresponds to the first episode of pediatric OCD, and Stage 2B corresponds to Multiple episodes or chronic pediatric OCD. This review focuses on early interventions and does not cover refractory OCD; therefore, stage 3 is not mentioned in detail in the article.

## Data Availability

Not applicable.

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
