# Peer review of "Early Identification and Intervention in Pediatric Obsessive-Compulsive Disorder"

_brainsci, 2023, doi:10.3390/brainsci13030399_

Round 1

Reviewer 1 Report

Primary prevention. The question is how the authors imagine the implantation of proposed techniques when they are reliable/available in practice. Do all children population undergo genome sequencing, volumetry in MRI, neuropsychological assessments? This is exciting approach but probably not possible to implement in real life in foreseeable future. Please give your opinion if the authors think I am not right.

Much more practical and not less interesting is secondary prevention. I miss the paragraph based on current knowledge that will address the following questions: Which children with OCS are at high risk to turn to OCD? Which young children with normal ritualized behavior are at risk to develop OCS and full-blown OCD in future? Does developmental normal ritualized behavior increase the risk for development of OCD later in lifetime. This is important regarding the rationale and different therapeutic strategies for secondary prevention – some children with OCS should be treated and some may not.

The similar statements are with regard to tertiary prevention. If pediatric OCD is related to high rates of remissions and often is episodic – which children are at risk to have OCD symptoms lasting longer and persisting into adulthood (current knowledge). In latter cases treatment should be probably more aggressive.  

Reviewer 2 Report

The early identification and intervention of pediatric obsessive-compulsive disorder (OCD) is imperative. This paper endeavors to critically examine the current impediments that hinder the promotion of early interventions for pediatric OCD, and to comprehensively review the various approaches that have the potential to aid in the early identification and intervention of pediatric OCD, which have been categorized according to three stages of prevention. This review is thorough in its scope. Nonetheless, the manuscript must undergo revision to enhance both its writing quality and content. Specifically, the authors should take into account the provided comments and suggestions.

Comments to the Authors: 

Why difficult? —— A Perspective on the Characteristics of Pediatric OCD

(1)The autoimmune and neuroinflammatory hypothesis have been suggested in varies neurodegenerative disorders, how the current findings to support or not support the autoimmune and neuroinflammatory hypothesis of OCD, requires further discussion. This is inadequate in the current version.

(2)The author should cite reference in “Given that the generality of immunomod- ulatory interventions and the biomarkers that can predict their efficacy remains unclear”(line 114-115).

Primary, Secondary and Tertiary prevention of pediatric OCD

The utilization of the CBT, as outlined in the 3.2.2 section, occurred without a prior definition being established (line 355).

Discussion

(1) It is unclear how pediatric OCD develops over time and how it is related to other mental health conditions. More research is needed to understand the natural course of the disorder and how it evolves with age, as it may contribute to the identifing high-risk children in early stage. I suggest the authors add a discussion of this point.

(2) Despite advances in our understanding of the neural mechanisms that contribute to OCD in adults, further neuroimaging studies are necessary in order to better comprehend the brain regions and potential abnormal patterns of functional connectivity involved in pediatric OCD. This will be important in informing the potential use of non-invasive neurostimulation techniques for early intervention. Further discussion of these topics may inspire and stimulate future research efforts in this area.

Round 2

Reviewer 2 Report

The author answered all my question in a good way. Congratulations.